# Up-Cycling Grape Pomace through Sourdough Fermentation: Characterization of Phenolic Compounds, Antioxidant Activity, and Anti-Inflammatory Potential

**DOI:** 10.3390/antiox12081521

**Published:** 2023-07-29

**Authors:** Andrea Torreggiani, Chiara Demarinis, Daniela Pinto, Angela Papale, Graziana Difonzo, Francesco Caponio, Erica Pontonio, Michela Verni, Carlo Giuseppe Rizzello

**Affiliations:** 1Department of Environmental Biology, “Sapienza” University of Rome, 00185 Rome, Italy; andrea.torreggiani@uniroma1.it (A.T.); carlogiuseppe.rizzello@uniroma1.it (C.G.R.); 2Department of Soil, Plant and Food Science, University of Bari Aldo Moro, 70126 Bari, Italy; chiara.demarinis@uniba.it (C.D.); graziana.difonzo@uniba.it (G.D.); francesco.caponio@uniba.it (F.C.); erica.pontonio@uniba.it (E.P.); 3Human Microbiome Advanced Project, 20129 Milan, Italy; dpinto@giulianipharma.com (D.P.); apapale@giulianipharma.com (A.P.)

**Keywords:** grape marc, lactic acid bacteria, fermentation, phenolic compounds, Caco2 cells

## Abstract

Despite its appealing composition, because it is rich in fibers and polyphenols, grape pomace, the major by-product of the wine industry, is still discarded or used for feed. This study aimed at exploiting grape pomace functional potential through fermentation with lactic acid bacteria (LAB). A systematic approach, including the progressively optimization of the grape pomace substrate, was used, evaluating pomace percentage, pH, and supplementation of nitrogen and carbon sources. When grape pomace was used at 10%, especially without pH correction, LAB cell viability decreased up to 2 log cycles. Hence, the percentage was lowered to 5 or 2.5% and supplementations with carbon and nitrogen sources, which are crucial for LAB metabolism, were considered aiming at obtaining a proper fermentation of the substrate. The optimization of the substrate enabled the comparison of strains performances and allowed the selection of the best performing strain (*Lactiplantibacillus plantarum* T0A10). A sourdough, containing 5% of grape pomace and fermented with the selected strain, showed high antioxidant activity on DPPH and ABTS radicals and anti-inflammatory potential on Caco2 cells. The anthocyanins profile of the grape pomace sourdough was also characterized, showing qualitative and quantitative differences before and after fermentation. Overall, the grape pomace sourdough showed promising applications as a functional ingredient in bread making.

## 1. Introduction

Wine industry by-products amount to roughly 5–7 million tons each year worldwide [1], with France, Italy, Spain, and the United States, in that order, being the biggest producers [2]. Within the wine brewing process, grape pomace is the major by-product and consists of peels, seeds, and a small amount of grape pulp, remaining after the wort is separated. Traditionally, part of the grape pomace generated is distilled to produce different local spirits; otherwise, it is dumped or used for animal feed or compost [3]. Nevertheless, grape pomace is rich in fibers and polyphenols; it is estimated that during wine brewing, roughly 60–70% of the total phenols, which include proanthocyanidins and a diversity of anthocyanin glycosides, remain in the pomace [4].

Such compounds could be extracted and recovered, although this would lead to more by-products. Hence, for its valorization, strategies that take into account a zero-waste approach, considering a qualitative and quantitative recovery optimization, should be preferred.

Several studies evaluated the effect of grape pomace addition to bread, pasta, cookies, muffins, dairy, and meat and fish products, yet all concluded that the fortification, especially if at high percentages, causes notable undesirable rheological and organoleptic changes [5]. Among the valorization strategies, fermentation, which is often used to improve the nutritional, functional, and technological properties of food and food by-products [6], might fit sustainability criteria. Few studies explored the possibility to ferment grape pomace with yeasts or edible fungi, yet very little can be found about the valorization strategies that use lactic acid bacteria (LAB) [7,8]. As a matter of fact, grape pomace, similar to grapes, can generate a hostile environment for the development of lactic acid bacteria, due to the low pH and high number of phenolic compounds, which are toxic for LAB [9,10]. Indeed, in winemaking, LAB species belonging to the genera *Pediococcus*, *Leuconostoc*, and the former *Lactobacillus* genus dominate grape must at the early stages of the fermentation, yet as the process goes on, most of them do not tolerate such conditions, and *Oenococcus* species mostly carry out the rest of the fermentation [9]. For this reason, it is imperative, when fermenting grape pomace, that an appropriate starter selection is performed, and the fermentation conditions optimized.

In this context, our study aimed at exploiting grape pomace functional potential through fermentation with LAB strains selected for their pro-technological and antioxidant features [11,12,13,14,15]. A systematic approach, which progressively guided us towards the proper strain and process conditions, was implemented to enhance pomace antioxidant and anti-inflammatory features with the prospect of its application as a functional ingredient in a sourdough bread.

## 2. Materials and Methods

### 2.1. Grape Pomace

The red grape pomace (*Vitis vinifera* L., cultivar Primitivo) used in this work was provided by a winery in Santeramo in Colle (Apulia, Southern Italy) following a seven-day maceration phase and collected after pressing. The grape pomace was dried at 70 °C for 60 min in a ventilated oven (Argolab-TCF120, Carpi, Italy) and finely ground with a laboratory mill Ika-Werke M20 (GMBH, and Co. KG, Staufen, Germany) to obtain a powder (grape pomace powder, GPP), further sieved with a 150 μm mesh. The proximate composition of grape pomace was the following: proteins, 11% of GPP dry matter (d.m.); lipids, 7% on d.m.; ashes, 10% on d.m.; and total dietary fiber, 42% on d.m.

Moisture was determined at 105 °C with a thermobalance MA35 (Sartorius, Gottinga, Germany); the activity water (a_w_) was determined with a Humimeter RH2 (Schaller Messtechnik, St. Ruprecht an der Raab, Austria).

### 2.2. Microrganisms

Eleven lactic acid bacteria strains belonging to the Culture Collection of the Department of Soil, Plant and Food Sciences, University of Bari Aldo Moro, were used as starters for GPP fermentation: *Furfurilactobacillus rossiae* T0A16 and LB5, *Lactiplantibacillus plantarum* T6B10, T0A10, 18S9, H18, H64, and LB1, *Pediococcus acidilactici* 10MM0, *Leuconostoc mesenteroides* 12MM1, and *P. pentosaceus* H22. LAB strains were singly cultivated in De Man, Rogosa and Sharpe (MRS, Oxoid, Basingstoke, Hampshire, UK) at 30 °C until the late exponential phase of growth was reached (ca. 10 h). Before the inoculum, cells were harvested by centrifugation (10,000× *g*, 10 min, 4 °C), washed twice in 50 mM phosphate buffer, pH 7.0, and re-suspended in tap water.

### 2.3. Starter Selection

#### 2.3.1. Grape Pomace-Derived Substrates

Aiming at investigating LAB performances, different conditions to render grape pomace to a suitable substrate for their growth were considered: (i) GPP/water ratio of 10%, 5%, and 2.5% (*w*/*v*); (ii) pH correction at pH 6.0 with food grade sodium bicarbonate (E500 as coded by European Food Safety Authority EFSA) (Solvay, Brussel, Belgium); (iii) supplementation with glucose (1% *w*/*v*) (Oxoid) and/or yeast extract (0.5% *w*/*v*) (Oxoid). Cells, collected as previously described, were resuspended in tap water, and used to singly inoculate the GPP substrates. Inoculum corresponded to 7 log cfu/mL; all fermentations were carried out at 30 °C for 24 h. For each condition considered, a not-inoculated control was prepared and analyzed before (Ct-T0) and after incubation at 30 °C for 24 h (Ct-T24).

#### 2.3.2. Growth and Kinetics of Acidification

With the aim of selecting the best adapting strain, LAB growth and acidification was monitored. A FiveEasy Plus (Mettler-Toledo, Columbus, OH, USA) pH-meter was used for monitoring the fermentation processes. TTA (total titratable acidity) was determined according to the official AACC method 02-31.01 [16] and expressed as the amount of 0.1 M NaOH required to adjust the pH of 10 g sample in sterile water to 8.3. LAB was enumerated using MRS agar (Oxoid) agar medium, supplemented with cycloheximide (0.1 g/L). Plates were incubated, under anaerobiosis (AnaeroGen and AnaeroJar, Oxoid), at 30 °C for 48 h.

The kinetics of acidification were modeled according to the Gompertz equation as modified by Zwietering [17]: y = k + A exp {−exp[(Vmaxe/A)(λ − t) + 1]}, where y is the acidification extent expressed as ΔpH at the time t, k is the initial level of the depend variable to be modeled, A is the difference pH between inoculation and stationary phase, Vmax is the maximum acidification rate, and λ is the length of the latency phase expressed in hours.

#### 2.3.3. In Vitro Antioxidant Activity

Methanolic extracts were obtained from GPP-derived substrates, as reported by Rizzello et al. [15], to determine the scavenging activity against the DPPH (2,2-diphenyl-1-picrylhydrazyl) radical [14]. The scavenging effect was expressed as shown in the following equation: DPPH scavenging activity (%) = [(blank absorbance − sample absorbance)/blank absorbance] × 100. The synthetic antioxidant butylated hydroxytoluene (BHT) was included as a reference (75 ppm) in the analysis.

### 2.4. Sourdough Fermentation and Characterization

Three type-II sourdoughs were produced by using type 0 wheat flour (Coop, Casalecchio di Reno, Italia) having the following proximal composition: moisture, 14.5% (*w*/*w*); proteins, 11.75% (*w*/*w*) on d.m.; carbohydrates, 83.4% (*w*/*w*) on d.m.; dietary fibers 2.9% (*w*/*w*) on d.m.; fat, 1.2% (*w*/*w*) on d.m.; ash, 0.7% (*w*/*w*) on d.m. GPP was mixed to wheat flour at percentages of 0, 2.5, and 5% (*w*/*w*), respectively, for sourdoughs SD0, SD2.5, and SD5.

Sourdoughs, having a dough yield (DY, dough weight × 100/flour weight) of 160, were produced using *L. plantarum* T0A10 as starter and fermented for 24 h at 30 °C.

The pH and TTA of the sourdoughs were determined before (t0) and after (t24) fermentation, as described above. Water/salt-soluble extracts (WSE) of the sourdoughs were prepared [18] and used to determine the content of total free amino acids (TFAA) by a Biochrom 30+ series Amino Acid Analyzer (Biochrom Ltd., Cambridge Science Park, Cambridge, UK) with a Li-cation-exchange column (4.6 × 200 mm internal diameter), as described by Verni et al. [19]. The WSE were also used to analyze lactic and acetic acids, respectively, with K-DLATE and K-ACET kits (Megazyme International Ireland Limited, Bray, Ireland). The quotient of fermentation (QF) was determined as the molar ratio between lactic and acetic acids. Radical scavenging activity on DPPH radical was determined on methanolic extracts, as described before. The evaluation of the scavenging activity against the ABTS+ radical (2,20-azino-di-(3-ethylbenzthiazoline sulfonate)) was evaluated using the CSO790 Kit (Sigma-Aldrich, Darmstadt, Germany), following the manufacturer’s instructions on both methanol and aqueous extracts. The scavenging activity was expressed as Trolox equivalents.

### 2.5. Analysis of Anthocyanins by UHPLC-DAD-MS/MS

Aiming at characterizing the phenolic profile of the SD5 sourdough, methanolic extracts were obtained before (SD5-T0) and after 24 h of fermentation (SD5-T24) at 30 °C. SD0 sourdough, not containing GPP, was analyzed as control.

A UHPLC Ultimate 3000RS Dionex interfaced by H-ESI II probe with a LTQ Velos pro linear ion trap mass spectrometer (Thermo Fisher Scientific, Waltham, MA, USA) was used for the analysis of anthocyanins compounds. The UHPLC system was composed by qua-ternary pump, autosampler, column compartment, and detector. The analytical separation was achieved as previously reported [20] with some modification. A Hypersil GOLD aQ C18 column was used (100 mm of length, 2.1 mm internal diameter and 1.9 μm of particle size), held at 30 °C and at a constant flow of 0.3 mL min^−1^ with water-formic acid (90:10 *v*/*v*) (solvent A) and acetonitrile-formic acid (99.9:0.1 *v*/*v*) (solvent B). The gradient program of solvent A was as follows: 0–20 min from 98% to 30%; 20–24 min isocratic at 30%. Then, equilibration was performed at the initial conditions for 9 min. The PDA detector was set to scan from 220 to 600 nm of wavelength managed by a 3D field.

The MS parameter conditions were as follows: capillary temperature 320 °C; source heater temperature 280 °C; nebulizer gas N2; sheath gas flow 33 psi; auxiliary gas flow 5 arbitrary units; S-Lens RF Level 60%. Data were acquired in positive ionization mode. Samples were analyzed with two methods: a full scan method from 100 to 1000 *m*/*z* and a data-dependent experiment to collect MS2 data. The data-dependent settings were a full scan from 200 to 1000 for positive ionization, activation level 500 counts, isolation width 2 Da, default charge state 2, and CID energy 35.

The samples were filtered using syringe filters (LLG Labware, Meckenheim, Germany) in RC by 0.22 µm before injection into the equipment. All data were acquired and processed using Xcalibur v.2 (Thermo Fisher Scientific, Waltham, MA, USA). The injection volume was 5 μL. Tentative identification of compounds was performed using mass spectra (MS2), λmax, and retention time according to the literature [21,22]. Quantitative analysis was performed according to the external standard method based on calibration curves obtained by injecting different concentrations of standard solutions (R2 = 0.9972). Specifically, the standard used was malvidin-3-*O*-glucoside, which was purchased from phyproof^®^ (PhytoLab, Dutendorfer, Germany).

### 2.6. Antioxidant Activity on Caco2 Cells

#### 2.6.1. Caco2 Cells Culture

Human colon carcinoma Caco2 cells (ICLC HTL97023) provided from the National Institute for Cancer Research of Genoa (Italy), were routinely cultured in Dulbecco’s modified Eagle’s medium (DMEM) GLUTAMAX medium supplemented with 10% (*v*/*v*) heat-inactivated fetal bovine serum (FBS), 1% (*v*/*v*) HEPES 1M, 1% (*v*/*v*) non-essential amino acids (Gibco), 1% L-glutamine 200 mM, penicillin (100 U/mL), and streptomycin (100 mg/mL) (Aurogene, Italy) and maintained in 25 cm^2^ culture flasks (BD Biosciences, Franklin Lakes, NJ, USA) at 37 °C in a 5% CO_2_ and 95% air-humidified atmosphere. Confluent cultures were split 1:3–1:6 every two days, after washing with PBS 1X (without Ca^2+^ and Mg^2+^), using Trypsin/EDTA and seed at 2–5 × 10^4^ cell/cm_2_, 37 °C, 5% CO_2_. Cell quantification was made through trypan blue assay.

Cell treatments were carried out by using freeze-dried methanolic extracts from SD5 (before and after 24 h of fermentation), resuspended in DMEM (10 mg/mL, stock solution), and sterilized through a 0.22 mm filter membrane (Millipore Corporation, Bedford, MA, USA).

#### 2.6.2. Citotoxicity

Cell viability was measured according to the MTT assay [23]. After 24 h seeding on a 96-well plate, 80% confluent Caco2 cells were exposed to 10 µg/mL of freeze dried methanolic extracts from SD5. The control was the basal medium. Plates were incubated at 37 °C, 5% CO_2_, for 24 h. After each treatment, the medium was aspirated and replaced with 100 μL per well of MTT solution. MTT was dissolved (5 mg/mL) in FBS and diluted 1:10 in the cell culture medium without phenol red. After 3 h of incubation, the basal medium was aspirated and 100 μL per well of DMSO were added to dissolve purple formazan product. The solution was shacked in the dark for 15 min at room temperature. The absorbance of the solutions was read at 570 nm in a microplate reader (BioTek Instruments Inc., Bad Friedrichshall, Germany). Each experiment was carried out in triplicate. Data were expressed as the mean percentage of viable cells compared to the culture in basal medium.

#### 2.6.3. RNA Extraction and Real-Time-PCR

After treatment with the extracts from SD5, the expression of *TNF-α* and *IL-1β* from Caco2 cells was investigated through RT-PCR. When ca. 80% confluence was reached, Caco2 cells were harvested with trypsin/EDTA, seeded, at a density of 1 × 10^6^ cells per well, into 12-well (Becton Dickinson France S.A., Meylan Cedex, France) plates and incubated at 37 °C, 5% CO_2_, for 24 h. Cells in DMEM GLUTAMAX medium and DMEM GLUTAMAX with lipopolysaccharide (LPS) (10 μg/mL) were used as the controls. Freeze-dried methanolic extracts from SD5 at a concentration of 10 μg/mL were added to 80% confluent Caco2 cells with the same LPS concentration (10 μg/mL) and incubated at 37 °C for 24 h. For quantitative real-time PCR (RT-PCR), total RNA from Caco2 cells was extracted using Tri Reagent (Sigma Aldrich), as described by Chomczynski and Mackey [24]. The cDNA was synthesized from 2 μg RNA template in a 20 μL reaction volume, using the PrimeScript RT-PCR kit (Takara, Japan).

The cDNA was amplified and detected by the Stratagene Mx3000P Real-Time PCR System (Agilent Technologies Italia S.p.A., Milan, Italy). The amplification of cDNA was conducted using the following Taqman gene expression assays: TNF-α Hs00174128_m1 (tumor necrosis factor alpha), IL-1β Hs01555410_m1 (interleukin-1-beta), and Hs999999 m1 (human glyceraldehyde-3-phosphate dehydrogenase, GAPDH). GAPDH was used as a housekeeping gene. PCR amplifications were carried out in a 20 µL of total volume. The mixture of reaction contained 10 µL of 2× Premix Ex Taq (Takara, Kusatsu, Japan), 1 µL of 20× TaqMan gene expression assay, 0.4 µL of RoX Reference Dye II (Takara, Japan), 4.6 µL of water, and 4 µL of DNA. PCR conditions were the following: 95 °C for 30 s followed by 40 cycles of 95 °C for 5 s and 60 °C for 20 s. PCR reactions were performed in duplicate using an MX3000p PCR machine (Stratagene, La Jolla, CA, USA). Analyses were carried out in triplicate. The average value of target gene was normalized using *GAPDH* gene and the relative quantification of the levels of gene expression was determined by comparing the Δ cycle threshold (ΔCt) value [25]. Results were expressed as percent ratio to cells only treated with LPS.

### 2.7. Breadmaking

Three experimental breads were manufactured by using the bread machine Ariete 132 Panexpress 750 (De Longhi Appliances Srl, Campi Bisenzio, Italy) and the type 0 flour above mentioned. Experimental breads were as follows: cY-B, a control bread, leavened with 2% *w*/*w* baker’s yeast; SD0-B, a control sourdough bread, containing 25% *w*/*w* of the SD0 sourdough, and leavened with 2% *w*/*w* baker’s yeast; SD5-B, a sourdough bread containing 25% *w*/*w* of the SD5 sourdough, and leavened with 2% *w*/*w* baker’s yeast.

All breads were obtained from doughs with DY 160 corresponding to a flour/water ratio of 62.5/37.5% (*w*/*w*), and all were added with commercial baker’s yeast (AB Mauri Italy S.p.a., Casteggio, Italia). Proofing was performed at 28 °C for 1.5 h and baking at 180 °C for 50 min. Recipes are reported in Appendix A.

### 2.8. Bread Characterization

#### 2.8.1. Biochemical and Nutritional Characterization

The analysis of pH, TTA, organic acids, and TFAA of the dough after proofing were carried out as reported above. The proximal composition and energy value of experimental breads were determined following the AACC approved methods [16]. In detail, protein (total nitrogen × 5.7), fats, ash, dietary fibers, and moisture were determined according to the 46-11A, 30–10.01, 08–01, 32-05.01, and 44-15A methods, respectively. Carbohydrates were calculated as the difference [100 − (proteins + lipids + ash + total dietary fibers + starch)].

#### 2.8.2. Technological Characterization

The dough leavening performance of the different samples was evaluated determining the volume increase of a 15 mL-dough placed in a graduated cylinder top, covered with a piece of Parafilm^®^ and allowed to ferment at 28 °C. Results were expressed as the difference between the initial and the final volume/min (∆V(mL)/min). Texture profile analysis was performed by using an FRTS-100N Texture Analyzer (Imada, Toyohashi, Japan) equipped with a 3 cm cylinder probe FR-HA-30J on boule-shaped loaves (200 g) stored for 2 h at room temperature after baking. The instrument settings were test speed 1 mm/s, 30% deformation of the sample, and two compression cycles, and the parameters evaluated were hardness, cohesiveness, springiness, and chewiness.

The chromaticity coordinates of the crust and crumb of the bread were obtained by a CS-10 colorimeter (CHN Spec Technology, Hangzhou, China) and reported as color difference, Δ*E* ∗ *ab*, calculated by the following equation:$$\Delta E*ab=\sqrt{{\left(\Delta L\right)}^{2}+{\left(\Delta a\right)}^{2}+{\left(\Delta b\right)}^{2}}$$
where Δ*L*, Δ*a*, and Δ*b* are the differences for *L*, *a**, and *b** values between sample and reference (a white ceramic plate having *L* = 92.2, *a** = 0.15, and *b** = 0.85).

#### 2.8.3. Sensory Analysis

Breads sensory analysis was performed by a trained panel group composed of 10 assessors (5 male and 5 females, mean age: 30 years, range: 25–54 years) with proven skills and previous experience in sensory evaluation of bread, pasta, and other cereal-based products. The sensory attributes, scored with a scale from 0 to 10 (with 10 being the highest score), were discussed with the assessors during the introductory 2 h-training session. Sensory attributes included: visual and tactile perception (crust and crumb color, elasticity, friability); taste (sweetness, bitterness, astringency, salty taste, herbaceous taste, acidic taste, overall flavor intensity); and scent perception (acidic odor). Bread slices (1.5 cm thick) were coded and served in a randomized order 4 h after baking. A glass of water was drunk by the panelists after each sample tasting.

### 2.9. Statistical Analysis

All the data were reported as the means of the data collected in three independent analyses. Data were subjected to one-way ANOVA; pair-comparison of treatment means was achieved by Tukey’s procedure at *p* < 0.05, using the statistical software Statistica 12.5 (StatSoft Inc., Tulsa, OK, USA). The results of anthocyanins quantification were subjected to one-way analysis of variance followed by Tukey’s procedure at *p* < 0.05, using OriginPro 2020 (OriginLab Corporation, Northampton, MA, USA).

## 3. Results

### 3.1. LAB Strain Selection

The GGP proximal composition (expressed as % on d.m.) was as follows: proteins, 12.35 ± 0.51; lipids 8.93 ± 0.21; dietary fibers, 53.14 ± 1.45; carbohydrates, 17.73 ± 0.14; ash, 7.84 ± 0.76. Moisture and a_w_ were, respectively, 5.76 ± 0.51% (*w*/*w*) and 0.3169 ± 0.0012.

Aiming at verifying the suitability of GPP as substrate for LAB growth, the 11 strains were inoculated in suspensions obtained with 10% GPP in water, with or without pH adjustment to 6.00 (initial pH of the GPP in water was 3.88 ± 0.02, while TTA was 10.4 ± 0.2 mL), with or without supplementations with glucose and/or yeast extract. No significant changes in pH and TTA were found for any of the conditions tested, while LAB cell density during the 24 h of incubation remained stable in conditions in which pH was corrected, while it decreased from 1 to 2 log cycles in all the others.

Assuming a strong inhibition of microbial growth linked to the abundance of polyphenols and pH, the test was repeated in a substrate containing lower GPP concentration, corresponding to 5% and 2.5% (*w*/*v*) in water. In substrates containing 5% *w*/*v* GPP, with the initial correction of pH to 6.00, a slight pH decrease (up to 0.5) was found only for three of the eleven strains inoculated (*L. plantarum* T0A10, H64, and *F. rossiae* T0A16). When the percentage of pomace in the substrate was reduced to 2.5%, the ΔpH value exceeded 0.5 units for all the strains inoculated.

In particular, acidification was intense when glucose and yeast extract were used simultaneously as supplements. Under these conditions, the decrease in pH was, for *L. plantarum* T0A10, *F. rossiae* T0A16, and *L. plantarum* H64, higher than 2.5 units.

Based on these results, the substrate containing 2.5% of GPP, adjusted to pH 6.00, and with added glucose and yeast extract was considered suitable for the subsequent comparison of the fermentative performances of the LAB strains.

The parameters obtained from the modeling of the acidification kinetics of the 11 strains are shown in Table 1. A, corresponding to the pH difference between the adaptation phase and the stationary phase, ranged from 2.878 and 2.361 (pH units), respectively for *F. rossiae* T0A16 and *L. plantarum* 1*8S9*. The strains having the highest A values were *F. rossiae* T0A16 and *L. plantarum* H64 and T0A10 (Table 1). The maximum acidification rate was between 0.475 and 0.311 ΔpH/h. In particular, *P. pentosaceus* H22, *F. rossiae* T0A16 and *L. plantarum* H64 showed Vmax values higher than 0.45. The adaptation phase was longer than 4 h for *L. plantarum* T6B10, H18, H64, *P. pentosaceus* H22, and *F. rossiae* T0A16 and shorter than 3.4 h for all others.

Cell densities reached by LAB strains in the pomace substrate are shown in Figure 1A. At the end of the 24 h incubation at 30 °C, the starters that presented the highest cell density were *L. plantarum* T0A10 (9.57 ± 0.12 log CFU/mL), *L. plantarum* LB1 (9.47 ± 0.11 log CFU/mL), and *L. plantarum* H18 (9.41 ± 0.10 log CFU/mL). Slightly lower values were found for *L. plantarum* H64, 18S9 and T6B10, and *Ln. mesenteroides* 12MM1. Significantly (*p* < 0.05) lower cell densities were found for all the others, with *P. acidilactici* 10MM0 showing the lowest final cell density (7.00 ± 0.01 log CFU/mL) (Figure 1A).

With the aim of investigating the effect of fermentation on the antioxidant activity of the substrate, the radical scavenging activity was determined for all fermented substrates (methanolic extracts), comparing the results to that of a non-inoculated control, before and after incubation in the same conditions of the inoculated samples. In particular, the antioxidant activity was expressed in terms of percentage of DPPH radical stabilized after 10 min of reaction. The samples with the highest antioxidant activity were those fermented by *L. plantarum* T6B10 (70.7 ± 0.9%), T0A10 (70.6 ± 2.5%), LB1 (69.4 ± 0.9%), and 18S9 (66.7 ± 1.2%), *Ln. mesenteroides* 12MM1 (65.5 ± 1.8%), and *L. rossiae* LB5 (65.3 ± 1.2%) (Figure 1B). The radical scavenging activity observed for the substrates fermented by *L. plantarum* T6B10, T0A10, and LB1 was comparable (*p* > 0.05) to that of the synthetic antioxidant BHT (75 ppm), also included in the analysis (71.21 ± 0.50%).

Markedly and significantly (*p* < 0.05) lower antioxidant activity was found for *P. acidilactici* 10MM0 (61.5 ± 2.1%), *L. plantarum* H64 (58.6 ± 2.0%), *F. rossiae* T0A16 (57.5 ± 1.8%), *P. pentosaceus* H22 (55.2 ± 2.3%), and *L. plantarum* H18 (49.20 ± 1.5%) (Figure 1B).

Based on the results obtained thus far, *L. plantarum* T0A10, showing a short λ phase (Table 1), high final cell density (Figure 1A) and high antioxidant activity (Figure 1B), was selected for further experiments and used as starter for sourdough fermentation of wheat flour containing GPP.

### 3.2. Sourdough Fermentation

Three type-II sourdoughs, differing only for the percentage of GPP added (0%, 2.5%, and 5% of flour weight), were produced and characterized. In all cases, intense acidification was observed over the 24 h of incubation. A final pH of ca. 3.60 was found for the three sourdoughs (without significant differences) (Table 2). However, the addition of GPP affected the pH of the dough before fermentation; indeed, both the initial pH and the ΔpH decreased as the percentage of added pomace increased (Table 2). The addition of GPP, by virtue of the high concentration of organic acids contained therein, also modified TTA values, which were higher for SD2.5 and SD5 compared to SD0. The selected *L. plantarum* strain grew by two logarithmic cycles in all three doughs, although the final cell density was significantly (*p* < 0.05) higher in SD0 (Table 2).

FAA was analyzed, before and at the end of the fermentation process, to evaluate the proteolytic activity of the strain. In SD0, the release of FAA was particularly consistent (+83% compared to t0), but in SD2.5 and SD5, the increases corresponded to 57% and 29%, respectively (Table 2).

Lactic acid concentration was significantly (*p* < 0.05) lower in SD2.5 and SD5 compared to SD0, and the same trend, albeit with markedly lower concentrations, was observed for acetic acid (Table 2). The fermentation quotient was lower than 5 for all three sourdoughs, but significantly (*p* < 0.05) higher for SD5 (4.97) compared to SD2.5 and SD0 (Table 2).

The antioxidant activity of the three sourdoughs was analyzed as radical scavenging activity on DPPH, before and after the fermentation (Table 2). The mere addition of GPP increased the antioxidant activity from about 7% (SD0) to 34 and 74% (for SD2.5 and SD5, respectively). Fermentation further increased the antioxidant activity in all three sourdoughs up to 40%, with values of radical scavenging activity of 48 and 95%, respectively, for SD2.5 and SD5 at the end of the 24 h of incubation (Table 2).

An additional assay aimed at quantifying the total antioxidant activity (on the ABTS radical) showed antioxidant activities of 0.48 and 1.02 mM Trolox equivalent for SD0 and SD5, respectively (Table 2). ABTS test also confirmed increases of the ABTS radical scavenging activity after sourdough fermentation, ranging from 60 to 70%. During sourdough fermentation, the pH kinetic was modeled (Table 2). The parameter *A* significantly (*p* < 0.05) decreased as the percentage of GPP increased (Table 2). The Vmax was lower in the two supplemented sourdoughs (although without significant differences between SD2.5 and SD5). The adaptation phase, increased together with the GPP supplementation entity, varying from 3.72 h in SD0 to 4.28 h in SD5 (Table 2).

### 3.3. Anthocyanins Identification and Quantification

Being the sourdough formulation exhibiting the most promising functional potential, SD5 was chosen for further characterization of the anthocyanins. Table 3 presents the concentration of individual anthocyanins identified in each sample based on their molecular ions and the corresponding anthocyanidin fragments produced in the MS^2^ experiment.

Overall, significant (*p* < 0.05) decreases, from 11 to 71%, were found in SD5-T24 compared to SD5-T0, for 6 out of 10 identified compounds. Whereas higher concentrations were found for carboxypyranomalvidin-3-*O*-glucoside and malvidin-3-*O*-trans-coumaroylglucoside after fermentation (74% and 17%, respectively). Another compound, malvidin 3,5-*O*-diglucoside, which was not found in SD5-T0, exceeded 20 mg/kg after fermentation.

### 3.4. Cytotoxicity and Anti-Inflammatory Effects

The MTT method was used as an indirect measure of cytotoxicity, through the determination of cell viability after treatments with the freeze-dried extracts from SD0 and SD5 (at concentrations of 0.01–0.1 mg/mL). After 24 h of incubation, none of the tested samples showed a significant (*p* > 0.05) cytotoxic effect on Caco2 cell line, compared to the control.

qRT-PCR was used to assess the anti-inflammatory potential of the samples, after cell treatment at 10 µg/mL. TNF-α and IL-1β, two cytokines involved in intestinal inflammation, were used as markers. Compared to positive control (medium + LPS), the treatment with SD5-T24 caused a marked and significant (*p* < 0.05) anti-inflammatory effect, with a decrease of the TNF-α gene expression of the 63% (Figure 2A), which was comparable (*p* < 0.05) to the expression observed in absence of LPS stimulation (Figure 2A). A lower, but also significant (*p* < 0.05) decrease (−17%) was also observed for SD5-T0 (Figure 2A). Compared to positive control (medium + LPS), only the SD5-T24 led to a significant (*p* < 0.05) decrease (−60%) of the *IL-1β* gene expression after 24 h of incubation (Figure 2B).

### 3.5. Breads

#### 3.5.1. Biochemical and Nutritional Characterization

For the production of the breads, three different doughs were obtained. While cY-B was produced only with the use of baker’s yeast, two others were added with 25% (of the total weight) of SD0 and SD5. During leavening (1.5 h at 28 °C), the increase in volume of the loaves (Table 4) was found to be similar (*p* > 0.05) for the three doughs. The final pH, on the other hand, was obviously lower for the loaves containing sourdough (4.59 and 4.48, respectively, for SD0-B and SD5-B). Accordingly, TTA had an opposite trend and was the lowest for cY-B (Table 4). The concentration of organic acids was affected by the sourdough presence. Indeed, lactic acid of SD0-B and SD5-B ranged from 23 to 24 mmol/kg, with no significant (*p* < 0.05) differences, while acetic acid was significantly higher in SD0-B. Organic acids were found only in traces in cY-B. The QF was lower in SD0-B compared to SD5-B (3.66 vs. 5.21). As expected, the concentration of FAA was higher in the sourdough breads than cY-B, obtained with baker’s yeast. The antioxidant activity on SD5-B corresponded to 46% and was markedly higher than the two bread formulations not containing GPP.

After cooking, the dough undergoes a drop in weight due to the evaporation of part of the water contained in the dough, hence the final moisture of the loaves was 27% (*w*/*w*).

Proximal composition of the breads was also determined (Appendix A). The main differences related to the presence of GPP (SD5-B) compared to the other two breads mainly concerned dietary fibers content, which was 30% higher in SD5-B compared to cY-B (2.72%). Consequentially, carbohydrates were slightly but significantly (*p* < 0.05) lower in SD5-B. The differences in the nutritional label did not determine significant variations in the energy value of the three experimental breads.

The analysis of free amino acids performed with the Biochrom 30 automatic amino acid analyser (Figure 3) showed that sourdough fermentation caused the increase of Ala, Val, GABA (γ-aminobutyric acid), Lys, Trp, Arg and Pro concentrations (SD5-B and SD0-B vs. cY-B), whereas the presence of pomace generated an increase in Cys and Tyr (SD5-B vs. cY-B).

#### 3.5.2. Technological Characterization

Breads were subjected to a Textural Profile Analysis (Table 5). The specific volume of the bread containing SD5 was significantly (*p* < 0.05) lower than that of bread obtained with only baker’s yeast (Table 5). The loaves had comparable (*p* > 0.05) hardness although the use of sourdough, especially SD5, gave lower cohesiveness to breads compared to baker’s yeast control. Unlike what is commonly found for fiber-enriched breads, the springiness parameter, which is related to elasticity, was the highest for SD5-B. Nevertheless, chewiness of SD5-B was higher compared to other bread. The addition of GPP caused considerable color differences, expressed overall by the parameter ΔE (Table 5). The colorimetric coordinates showed a significant (*p* < 0.05) decrease in brightness (L) for both the crust and the crumb and, for the crust, a value of the parameter “a” (green/red index) considerably higher compared to that observed for the other two breads (Table 5).

### 3.6. Sensory Profile

The sensory profile of the breads was assessed through a panel test. The average scores obtained from the evaluations of 10 assessors showed that the most evident differences of the SD5-B compared to the breads not containing GPP were related to crust and crumb color, which was markedly more intense (Figure 4).

The use of sourdough in SD0-B and SD5-B moreover conferred perceivable acidic taste and odor compared to cY-B (Figure 4). The friability of SD5-B was judged similar to the baker’s yeast control, despite the higher presence of fibers. Astringency and herbaceous flavors, attributes included because typical of grape pomace, were clearly recognized by panelists, but not perceived as defect. Among the three breads, SD5-B received the highest score for the overall flavor intensity.

## 4. Discussion

As one of the major by-products of the wine industry, grape pomace valorization strategies that consider both quantitative and qualitative recovery optimizations, should be sought, especially in world-leading countries for wine production such as Italy. Exploiting grape pomace potential could be beneficial not only for the sustainability of the whole food system, but also to boost the functionality of the product made thereof. Several studies have demonstrated the correlation between grape pomace phenolics, particularly procyanidins, and antioxidant, antimutagenic, and anticarcinogenic activities [26]. Nevertheless, despite their health-promoting properties, it is well known that phenolic compounds can be ester bound or trapped within proteins or polysaccharides, on cell walls of food matrices [27]. Being bound to dietary fibers, the fraction potentially bio-accessible and absorbed in the gastro-intestinal tract can change [27]. Over the years, numerous studies have found that bioprocessing treatments, including fermentation, can modify the free/bound phenolics ratio. Indeed, several enzymes, among which feruloyl esterases, glucosidases, and tannases, have been described in LAB, mostly of the former *Lactobacillus* species, in fungi of the genera *Aspergillus* and *Penicillium*, as well as in *Saccharomyces cerevisiae* [28].

Based on the above considerations, this study aimed at exploiting grape pomace functional potential through fermentation with 11 lactic acid bacteria strains previously found to enhance the nutritional features of the matrices from which they were isolated [11,12,13,14,15]. Regardless, an obstacle that should be considered when fermenting grape pomace is its suitability as growth substrate. This is why a first experimental phase was aimed at identifying the percentage of pomace which, resuspended in water, would allow LAB to grow and acidify the medium. In fact, it was expected that, due to (i) the conspicuous removal of soluble nutrients (fermentable sugars and nitrogenous substances) during the winemaking process, (ii) the low pH [9], and (iii) the high concentration of polyphenols, having marked antimicrobial activity [10], the substrate was not optimal for microbial growth. Although to grape pomace polyphenols have been ascribed several functional properties, they can directly inhibit LAB growth by (i) interacting with cell membrane, thus causing membrane permeabilization and leakage of intracellular components, (ii) inhibiting enzymes essential for their metabolism, or (iii) chelating metal ions and nutrients [29]. For these reasons, when GPP was used at 10%, especially without pH correction, LAB cell density decreased up to 2 log cycles in 24 h. Hence, GPP percentage was lowered to 5 or 2.5% and supplementations with carbon and nitrogen sources, which are crucial for LAB metabolism, acting as an energy source and playing an important role in the growth [30], were considered. The optimization of the substrate enabled the comparison of strains performances and allowed the selection of the best performing strain (*L. plantarum* T0A10) based on growth, acidification, and ability to increase the DPPH radical scavenging activity. Indeed, among lactic acid bacteria, *L. plantarum* is the most utilized species for plant substrate fermentation due to its remarkable, although strain-dependent, metabolic activity against phenolic compounds [29]. Such activities allow the detoxification of phenolic compounds into metabolites having less antimicrobial activity and often higher antioxidant potential [28,29] justifying the best adaptation of the selected strain and the increased DPPH radical scavenging activity (up to 20%) compared to unfermented GPP (Figure 1B).

Hence, further experiments, aimed at including GPP in sourdough production, only involved *L. plantarum* T0A10. Based on the substrate optimization, only 2.5% and 5% of GPP supplementation were considered, whereas the modification of the pH was naturally provided by the buffering effect of the flour (Table 2). Despite all, LAB cell density, ΔpH, concentrations of organic acids, and release of FAA suggested an inhibitory effect of the pomace on the microorganism, albeit rather limited. Yet, considerable (*p* < 0.05) increases of the DPPH and ABTS radical scavenging activity were observed in sourdoughs (SD2.5 and SD5) proportional to the percentage of addition. Indeed, as expected, the sole addition of GPP enhanced the in vitro antioxidant activity (SD2.5-T0 and SD5-T0 compared to SD0), yet fermentation with *L. plantarum* T0A10 determined a further and significant increase (Table 2), confirming the results obtained in the substrate optimization process. Similar results in terms of antioxidant activity were obtained when grape pomace was fermented with edible fungi *Phanerochaete chrysosporium* and *Trametes gibbosa* [24]. Being the percentage providing the most functional potential, the sourdough containing the highest GPP content (SD5) was chosen for further characterization.

Studies have shown that anthocyanins and their glycosides are the most abundant phenolic compounds in grape pomace [4], which is why the characterization of the phenolic profile focused on this class of compounds. Overall, the total amount of anthocyanins did not differ before and after fermentation, yet differences in the quantities of some of the identified compounds were found. In particular, malvidin 3,5-*O*-diglucoside, which was not detected in SD5-T0, was found in SD5-T24. As explained above, it is likely that in GPP, malvidin 3,5-*O*-diglucoside was part of the bound fraction of anthocyanins, and as a consequence of fermentation, either due to the pH or the enzymatic activity [28,29,31] of *L. plantarum* T0A10, it was released in free form. Although malvidin 3,5-*O*-diglucoside is not commonly found in *V. vinifera*, traces of it were detected when grape pomace was inoculated with *Lacticaseibacillus casei* [32], suggesting a possible release as a consequence of fermentation. The same could be said for carboxypyranomalvidin-3-*O*-glucoside and malvidin-3-*O*-trans-coumaroylglucoside, which also increased with the fermentation. It should also be highlighted that several factors, among which the concentration and the structure of these compounds, as well as the pH of the environment are directly related to anthocyanins antioxidant capacity [31]. Indeed, it was observed that the radical scavenging activity of anthocyanins extracts significantly increases at acidic pH, which results in a more stable form of reduced anthocyanins by free radicals [33], explaining why, despite the total amount not differing between SD5-T0 and SD5-T24, higher DPPH and ABTS radical scavenging activity was found after fermentation (Table 2). Moreover, it was found by Han et al. [34] that malvidin-3-*O*-trans-coumaroylglucoside, which was higher in SD5-T24, has higher antioxidant capacity compared to other anthocyanins [34].

Along with substrate optimization and starter selection, it is worth underlining the importance of the methods used for the determination of food functional properties. In fact, when *L. plantarum* PU1 or *Bifidobacterium breve* 15A were used to ferment grape marc no significant variation of the antioxidant activity, estimated by ABTS, was observed compared to the unfermented control, whereas high inhibition of the oxidation of linoleic acid as well as protective effect against Caco2 cell oxidative stress was observed [7]. Thus, since in vitro assays can only be considered predictive tools of the functional activity in vivo and testing substances directly on animals or human is not an easy approach, ex vivo assays offer the right compromise to better understand food functional properties. In our study, the MTT assay on the Caco2 cell line was used to determine the cytotoxicity of SD5, before and after fermentation, and none was found. Conversely, the expression of *TNF-α* and *IL-1β* from Caco2 cells was investigated through RT-PCR after treatment with the extracts from SD0 and SD5. In vitro and in vivo antioxidant and anti-inflammatory properties of grape pomace, mainly ascribed to its phenolic compounds, have been already reported (for a review, see Bocsan et al. [35]). Yet, in our study, fermentation led to a further down-regulation of the expression of the pro-inflammatory proteins, such as cytokines (*IL-1β* and *TNF-α*), showing a significant anti-inflammatory effect of SD5-T24 compared to the other sourdoughs (before fermentation or without GPP addition). This could be due to a series of factors, among which the liberation into a free form of GPP anthocyanins which were not in the SD5-T0 and the effect of the pH. Moreover, in the conditions of this study, only anthocyanins were detected, whereas, even though limited, a contribution of the phenolic compounds provided by the wheat flour used for sourdough, as well as the effect of *L. plantarum* metabolism on them, cannot be excluded. Indeed, a series of hydroxycinnamic and hydroxybenzoic acids constitute most of the free phenolic compounds of the flour [36] and lactic acid bacteria are known for the ability to decarboxylate phenolic acids to the corresponding phenol or vinyl derivatives or hydrogenated them through phenolic acid reductases [28]. Metabolites of phenolic acids conversion, compared to their precursors, have reduced antimicrobial activity, but higher biological activities than their precursors [28], thus explaining the overall higher functional features of SD5-T24 compared to SD5-T0.

The functionality and health benefits of foods, especially fermented ones, are becoming increasingly important for consumers and in tourn food industries, which is why more research should focus on these aspects, particularly if a step towards the overall sustainability of the process is to be taken [29]. Therefore, SD5 was used as ingredient in bread making. In terms of nutritional composition, the main difference among breads was the higher fiber content in SD5-B compared to SD0-B and Cy-B. It should be noted that the increase in fiber is not only due to the addition of pomace, but we can hypothesize a contribution from resistant starch, which is generated following the biological acidification [37]. During sourdough fermentation, although negatively affected by GPP addition, a copious increase of GABA (up to 50-fold) was found compared to Cy-B. Such increase is of particular interest since γ-aminobutyric acid, produced by LAB as response mechanism to the low pH, is a functional compound with hypotensive, diuretic, and tranquilizing effects [38] which could further enhance bread functionality. The addition of GPP in the bread formulation, as expected, slightly impacted its technological properties; nevertheless, due to the small fortification and sourdough fermentation, known for its ability to modify rheology attributes [39], such effects were mitigated. These rheological attributes mainly rely on the physicochemical changes of the protein network, as a consequence of the acidic pH, which facilitate the larger dough expansion during fermentation [39]. Finally, the addition of SD5 to bread broadened its aroma profile with herbaceous and acid tastes due to the GPP itself and the sourdough fermentation, respectively. Overall, higher and pleasant flavor intensity was observed by panelists for SD5-B, which could be due to a variety of volatile (e.g., alcohols, acids, ketones, hydrocarbons, aldehydes, and esters) and non-volatile (e.g., amino acids) compounds developed during GPP fermentation [29]. Indeed, among non-volatile compounds proline and aspartic acid, which were all significantly higher (up to 11-fold) in SD5-B compared to Cy-B, are the ones that affect flavor the most [29].

## 5. Conclusions

This study demonstrated that lactic acid bacteria, specifically selected to ferment grape pomace, the major by-product of the wine industry, can be used to exploit its functional potential, increasing its antioxidant and anti-inflammatory features. The combination of sourdough fermentation and grape pomace proved to be an interesting ingredient for sourdough bread, with prospects of application that can boost both the sustainability of the food system and the intake of healthier and functional foods.

## Figures and Tables

**Figure 1 antioxidants-12-01521-f001:**
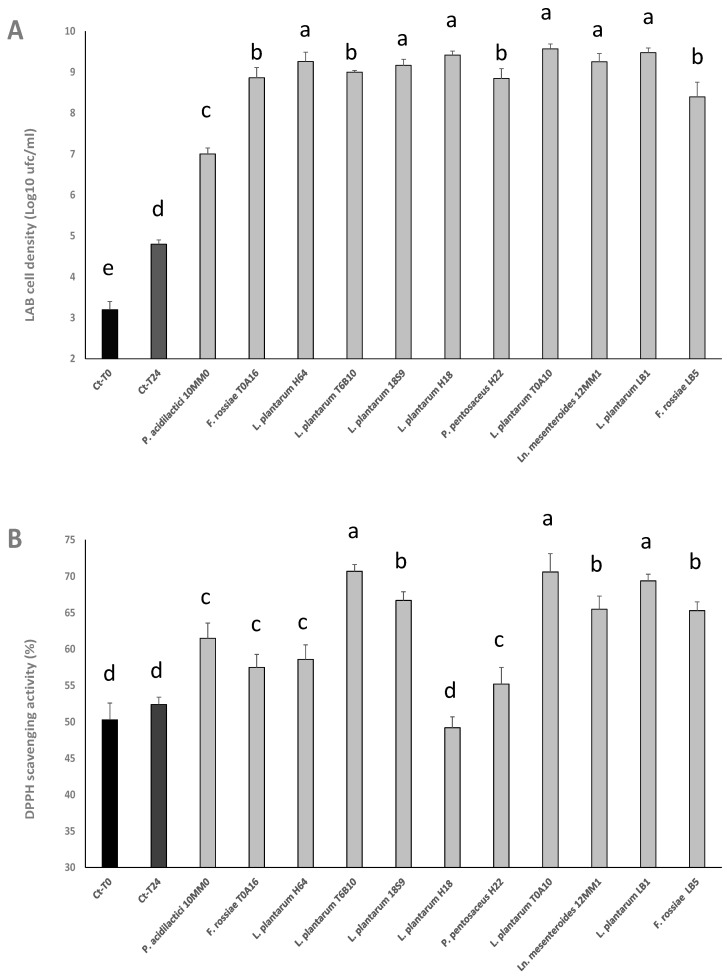
LAB cell density (**A**) and radical scavenging activity on DPPH radical (**B**) in grape pomace substrate (GPP 2.5% *w*/*v*, glucose 1% *w*/*v*; yeast extract 0.5% *w*/*v*, pH 6.0) inoculated (7 log cfu/mL) with *Furfurilactobacillus rossiae* T0A16 and LB5, *Lactiplantibacillus plantarum* T6B10, T0A10, 18S9, H18, H64, and LB1, *Pediococcus acidilactici* 10MM0, *Leuconostoc mesenteroides* 12MM1, and *P. pentosaceus* H22, and fermented at 30 °C for 24 h. A not-inoculated control was included and analyzed before (Ct-T0) and after incubation (Ct-T24). Data ± are the means of three independent analyses. ^a–d^ Values with different superscript letters differ significantly (*p* < 0.05). Bars of standard deviations are also represented.

**Figure 2 antioxidants-12-01521-f002:**
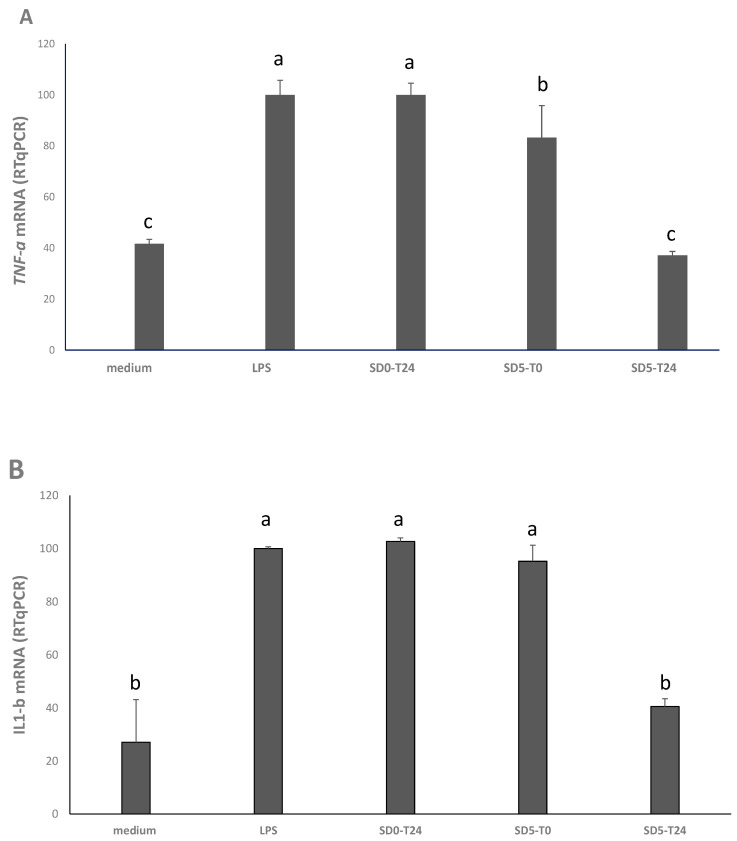
Expression of the tumor necrosis factor alpha (TNF-α) (panel **A**) and interleukin1 beta (Il-1β) (panel **B**) genes in Caco2 cells as determined by RT-PCR. Caco2 cells were treated at 37 °C for 24 h, under 5% CO_2_, with: basal medium (negative control); basal medium with LPS (10 μg/mL) (positive control, LPS); and 10 μg/mL of the freeze-dried extracts from SD0 and SD5, before (T0) and after (T24) sourdough fermentation. Results are expressed as percent ratio to LPS treatment. Values are means of two experiments in triplicate. ^a–c^ Values with different superscript letters differ significantly (*p* < 0.05). Bars of standard deviations are also represented.

**Figure 3 antioxidants-12-01521-f003:**
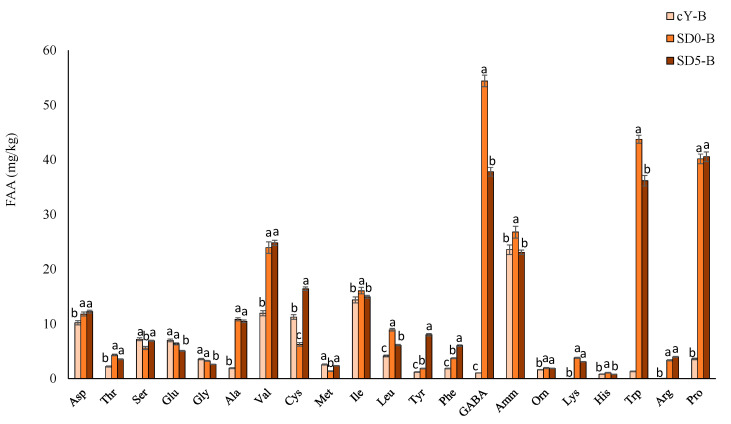
Free amino acids (mg/kg) in experimental breads. cY-B, a control bread, leavened with 2% *w*/*w* baker’s yeast; SD0-B, control sourdough bread, containing 25% *w*/*w* of the SD0 sourdough, and leavened with 2% *w*/*w* baker’s yeast; SD5-B, a sourdough bread containing 25% *w*/*w* of the SD5 sourdough, and leavened with 2% *w*/*w* baker’s yeast. Data are the means of three independent analyses. ^a–c^ Values with different superscript letters differ significantly (*p* < 0.05). Bars of standard deviations are also represented.

**Figure 4 antioxidants-12-01521-f004:**
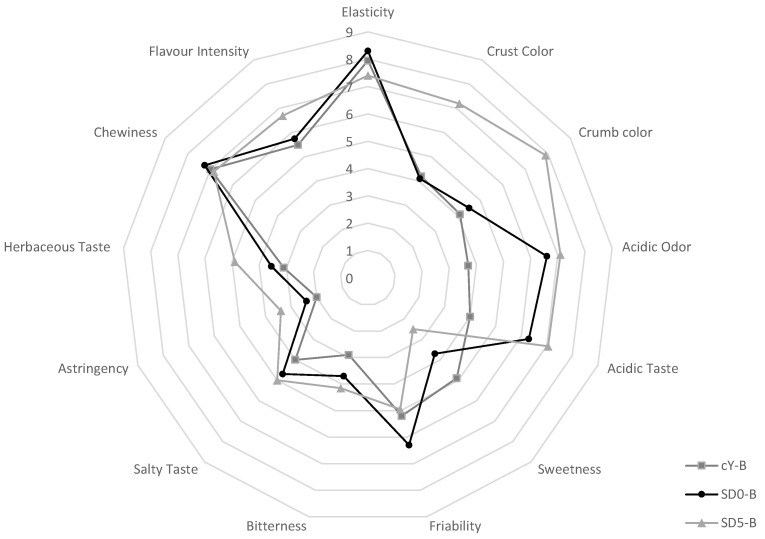
Spider web chart of the results obtained in the sensory analysis of the experimental breads. cY-B, control bread, leavened with 2% *w*/*w* baker’s yeast; SD0-B, control sourdough bread, containing 25% *w*/*w* of the SD0 sourdough, and leavened with 2% *w*/*w* baker’s yeast; SD5-B, a sourdough bread containing 25% *w*/*w* of the SD5 sourdough, and leavened with 2% *w*/*w* baker’s yeast.

**Table 1 antioxidants-12-01521-t001:** Parameters of the acidification kinetics. A, difference of the pH between inoculation and the stationary phase (ΔpH); Vmax, the maximum acidification rate (ΔpH/h); λ, length of the latency phase (h).

*Strain*	A	Vmax	λ
*L. plantarum* T0A10	2.83 ± 0.08 ^a^	0.41 ± 0.02 ^b^	3.39 ± 0.06 ^d^
*Ln. mesenteroides* 12MM1	2.47 ± 0.05 ^b^	0.41 ± 0.03 ^b^	3.46 ± 0.05 ^d^
*L. plantarum* LB1	2.41 ± 0.06 ^c^	0.39 ± 0.01 ^b^	3.31 ± 0.02 ^d^
*L. plantarum* T6B10	2.40 ± 0.09 ^c^	0.40 ± 0.03 ^b^	4.39 ± 0.05 ^c^
*L. plantarum* 18S9	2.36 ± 0.08 ^d^	0.31 ± 0.02 ^c^	3.31 ± 0.02 ^d^
*L. plantarum* H18	2.46 ± 0.05 ^b^	0.44 ± 0.02 ^a^	5.19 ± 0.07 ^a^
*P. pentosaceus* H22	2.44 ± 0.02 ^b^	0.47 ± 0.03 ^a^	5.44 ± 0.07 ^a^
*F. rossiae* T0A16	2.88 ± 0.02 ^a^	0.46 ± 0.03 ^a^	4.94 ± 0.02 ^b^
*L. plantarum* H64	2.87 ± 0.04 ^a^	0.47 ± 0.02 ^a^	4.92 ± 0.03 ^b^

The data are the means of three independent experiments ± standard deviations (n = 3). ^a–d^ Values in the same column with different superscript letters differ significantly (*p* < 0.05).

**Table 2 antioxidants-12-01521-t002:** Characterization of the type II sourdoughs inoculated with *L. plantarum* T0A10 (7 log cfu/g) and fermented at 30 °C for 24 h. GPP was mixed to the wheat flour at percentages of 0, 2.5, and 5% (*w*/*w*), respectively, for sourdoughs SD0, SD2.5, and SD5. Dough yield of the sourdoughs was 160.

	SD0		SD2.5		SD5	
	t0	tf	t0	tf	t0	tf
**pH**	5.69 ± 0.04 ^a^	3.61 ± 0.03 ^d^	5.12 ± 0.02 ^b^	3.63 ± 0.01 ^d^	4.73 ± 0.01 ^c^	3.59 ± 0.02 ^d^
**TTA**	1.82 ± 0.01 ^e^	10.2 ± 0.18 ^b^	4.21 ± 0.03 ^d^	12.8 ± 0.26 ^a^	6.22 ± 0.05 ^c^	14.03 ± 0.32 ^a^
**Lactic acid bacteria (Log ufc/g)**	2.03 ± 0.10 ^c^	9.96 ± 0.09 ^a^	2.74 ± 0.30 ^c^	9.24 ± 0.07 ^b^	2.80 ± 0.25 ^c^	9.05 ± 0.13 ^b^
**Lactic acid (mmol/kg)**	nd	90.41 ± 2.31 ^a^	nd	85.23 ± 4.01 ^b^	nd	80.21 ± 3.87 ^c^
**Acetic acid (mmol/kg)**	nd	24.64 ± 1.05 ^a^	1.22 ± 0.07 ^b^	19.21 ± 2.27 ^b^	1.98 ± 0.10 ^b^	16.12 ± 1.54 ^c^
**QF**	-	3.67 ± 0.16 ^c^	-	4.43 ± 0.29 ^b^	-	4.97 ± 0.17 ^a^
**Total free amino acids (mg/kg)**	314 ± 12 ^d^	576 ± 32 ^a^	309 ± 16 ^d^	485 ± 27 ^b^	290 ± 10 ^e^	375 ± 21 ^c^
**DPPH Radical Scavenging Activity (%)**	7.0 ± 0.3 ^f^	13.1 ± 0.7 ^e^	34.3 ± 0.9 ^d^	58.4 ± 0.8 ^c^	74.1 ± 1.3 ^b^	95.2 ± 1.6 ^a^
**ABTS Radical scavenging (mM Trolox eq)**	0.11 ± 0.03 ^d^	0.18 ± 0.05 ^d^	0.30 ± 0.08 ^c^	0.48 ± 0.15 ^b^	0.62 ± 0.10 ^b^	1.02 ± 0.03 ^a^
	*Kinetics of acidification parameters*					
**A**	2.04 ± 0.12 ^a^		1.48 ± 0.07 ^b^		1.17 ± 0.04 ^c^	
**Vmax (ΔpH/Δh)**	0.42 ± 0.02 ^a^		0.25 ± 0.01 ^b^		0.22 ± 0.01 ^b^	
**λ (h)**	3.72 ± 0.18 ^b^		3.89 ± 0.13 ^b^		4.28 ± 0.10 ^a^	

The data are the means of three independent analysis ± standard deviations (n = 3). ^a–e^ Values in the same row with different superscript letters differ significantly (*p* < 0.05).

**Table 3 antioxidants-12-01521-t003:** Retention time, concentration (mg/kg), mass spectral, and UV-Vis data of anthocyanins identified in the type II sourdoughs with 5% (*w*/*w*) of GPP, before (SD5-T0) and after (SD5-T24) fermentation with *L. plantarum* T0A10.

Peak	RT (min)	Concentration (mg/kg)		MH^+^ (*m*/*z*)	Fragments	λ_max_	Anthocyanins
		SD5-T0	SD5-T24				
**1.**	6.68	41.7 ± 1.6 ^a^	28.4 ± 0.9 ^b^	479	317	278,524	Petunidin 3-*O*-glucoside
**2.**	7.16	-	23.3 ± 0.1	655	331,493	276,530	Malvidin 3,5-*O*-diglucoside
**3.**	7.37	40.3 ± 3.7 ^a^	11.8 ± 0.8 ^b^	463	301	282,522	Peonidin 3-*O*-glucoside
**4.**	7.66	847.5 ± 13.5 ^a^	755.3 ± 22.0 ^b^	493	331	258,528	Malvidin 3-*O*-glucoside
**5.**	8.12	12.6 ± 0.9 ^b^	21.9 ± 0.6 ^a^	561	399	280,512	Carboxypyranomalvidin-3-*O*-glucoside
**6.**	9.49	31.4 ± 2.3 ^a^	21.0 ± 1.5 ^b^	535	331	280,522	Malvidin-3-*O*-acetylglucoside
**7.**	9.99	30.2 ± 1.5 ^a^	25.6 ± 0.3 ^b^	655	331	280,530	Malvidin-3-*O*-caffeoylglucoside
**8.**	10.13	34.9 ± 0.01 ^a^	32.3 ± 1.4 ^a^	625	317	282,530	Petunidin-3-*O*-coumaroylglucoside
**9.**	10.68	41.5 ± 0.2 ^a^	34.7 ± 1.3 ^b^	609	301	282,522	Peonidin 3-*O*-coumaroylglucoside
**10.**	10.84	351.3 ± 13.4 ^b^	410.8 ± 10.1 ^a^	639	331	282,534	Malvidin-3-*O*-trans-coumaroylglucoside

The data are the means of two independent analysis ± standard deviations (n = 3). ^a–b^ Values in the same row with different superscript letters differ significantly (*p* < 0.05).

**Table 4 antioxidants-12-01521-t004:** Bread characterization data: cY-B, a control bread, leavened with 2% *w*/*w* baker’s yeast; SD0-B, control sourdough bread, containing 25% *w*/*w* of the SD0 sourdough, and leavened with 2% *w*/*w* baker’s yeast; SD5-B, a sourdough bread containing 25% *w*/*w* of the SD5 sourdough, and leavened with 2% *w*/*w* baker’s yeast.

	cY-B	SD0-B	SD5-B
** *Volume increase (mL/min)* **	0.212 ± 0.013 ^a^	0.198 ± 0.009 ^a^	0.205 ± 0.008 ^a^
** *pH* **	5.61 ± 0.02 ^a^	4.59 ± 0.01 ^b^	4.48 ± 0.02 ^b^
** *TTA (mL NaOH 0.1 M)* **	2.5 ± 0.13 ^b^	4.82 ± 0.24 ^a^	5.10 ± 0.16 ^a^
** *Lactic acid (mmol/kg)* **	1.32 ± 0.11 ^b^	23.13 ± 0.49 ^a^	24.18 ± 0.62 ^a^
** *Acetic acid (mmol/kg)* **	1.40 ± 0.21 ^c^	6.32 ± 0.28 ^a^	4.64 ± 0.36 ^b^
** *QF* **	-	3.66 ± 0.19 ^b^	5.21 ± 0.14 ^a^
** *Total Free Amino acid (mg/kg)* **	111 ± 4 ^b^	279 ± 8 ^a^	267 ± 6 ^a^
** *DPPH Radical Scavenging Activity (%)* **	7.3 ± 0.2 ^c^	12.0 ± 1.3 ^b^	46.2 ± 2.3 ^a^

The data are the means of three independent analysis ± standard deviations (n = 3). ^a–c^ Values in the same row with different superscript letters differ significantly (*p* < 0.05).

**Table 5 antioxidants-12-01521-t005:** Volume, textural and color features of the experimental breads. cY-B, control bread, leavened with 2% *w*/*w* baker’s yeast; SD0-B, control sourdough bread, containing 25% *w*/*w* of the SD0 sourdough, and leavened with 2% *w*/*w* baker’s yeast; SD5-B, a sourdough bread containing 25% *w*/*w* of the SD5 sourdough, and leavened with 2% *w*/*w* baker’s yeast.

	cY-B	SD0-B	SD5-B
**Specific volume (cm^3^/g)**	3.12 ± 0.06 ^a^	3.06 ± 0.03 ^ab^	2.92 ± 0.04 ^b^
**Hardness (N)**	52.51 ± 1.20 ^a^	51.76 ± 2.05 ^a^	52.76 ± 0.14 ^a^
**Cohesiveness**	0.57 ± 0.00 ^a^	0.46 ± 0.05 ^b^	0.35 ± 0.02 ^c^
**Springiness**	0.84 ± 0.02 ^b^	0.84 ± 0.01 ^b^	1.15 ± 0.10 ^a^
**Chewiness (N)**	27.06 ± 3.37 ^b^	19.95 ± 0.95 ^c^	44.30 ± 0.07 ^a^
Crust color			
**L**	62.86 ± 1.43 ^a^	57.51 ± 0.34 ^a^	46.15 ± 2.79 ^b^
**a**	−1.61 ± 0.24 ^b^	−1.90 ± 0.34 ^b^	1.20 ± 0.27 ^a^
**b**	18.93 ± 0.38 ^a^	16.54 ± 1.57 ^a^	12.18 ± 0.89 ^b^
**ΔE**	33.99 ± 1.34 ^b^	38.07 ± 3.56 ^b^	47.97 ± 2.61 ^a^
Crumb color			
**L**	60.95 ± 3.63 ^a^	56.61 ± 0.20 ^b^	45.37 ± 2.05 ^c^
**a**	−4.3 ± 0.52 ^a^	−3.63 ±0.20 ^a^	−0.15 ± 0.15 ^b^
**b**	13.54 ± 0.4 ^a^	12.59 ± 0.67 ^a^	8.75 ± 0.14 ^b^
**ΔE**	34.02 ± 3.47 ^c^	37.88 ± 0.11 ^b^	48.24 ± 2.02 ^a^

The data are the means of three independent analysis ± standard deviations (n = 3). ^a–c^ Values in the same row with different superscript letters differ significantly (*p* < 0.05).

## Data Availability

The data presented in this study are available upon request from the corresponding author.

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
