# Peer review of "Up-Cycling Grape Pomace through Sourdough Fermentation: Characterization of Phenolic Compounds, Antioxidant Activity, and Anti-Inflammatory Potential"

_antioxidants, 2023, doi:10.3390/antiox12081521_

Round 1
Reviewer 1 Report
The work is interesting, it was well conducted experimentally and it is well written. Small details need to be corrected before publication.
It is important to describe the grape cultivar used in the wine, and which generated the residue (Section 2.1). Possibly it is Vitis vinifera L. In this case, in Table 3, the presence of malvidin 3,5 dilglucoside does not make sense since this pigment is typical of Vitis labrusca or hybridus grapes. This fact needs to be discussed in the work.
In table 2, the % inhibition of DPPH in some treatments is less than 20%. DPPH does not have adequate linearity outside the 20-80% range. At values lower than 20% it should not be detected. Please see: Brand-Williams, Cuvelier & Berset (1995). Use of a free radical method to evaluate antioxidant activity. https://doi.org/10.1016/S0023-6438(95)80008-5
Author Response
The work is interesting, it was well conducted experimentally and it is well written. Small details need to be corrected before publication.
The authors thank the reviewer for the comment.
It is important to describe the grape cultivar used in the wine, and which generated the residue (Section 2.1). Possibly it is Vitis vinifera L. In this case, in Table 3, the presence of malvidin 3,5 dilglucoside does not make sense since this pigment is typical of Vitis labrusca or hybridus grapes. This fact needs to be discussed in the work.
Paragraph 2.1 was implemented with info on grape cultivar (Line 74).
Reviewer is correct, malvidin 3,5-O-diglucoside is not usually found in grape pomace from Vitis vinifera, indeed, it was not detected when the SD5 was analyzed before fermentation. On the contrary it was found in the sourdough after fermentation, which is why we hypothesized it was present in a bound form, which thanks to the acidification and the metabolic activity of L. plantarum used for fermentation it was liberated into a free form. As a matter of fact, although unusual as the reviewer suggested, there are reports of grape pomace fermented with lactic acid bacteria, in which traces of malvidin 3,5-O-diglucoside were detected (Anghel et al., 2023. Dried grape pomace with lactic acid bacteria as a potential source for probiotic and antidiabetic value-added powders. Food Chem). This aspect was implemented in the discussion (Lines 616-618).
In table 2, the % inhibition of DPPH in some treatments is less than 20%. DPPH does not have adequate linearity outside the 20-80% range. At values lower than 20% it should not be detected. Please see: Brand-Williams, Cuvelier & Berset (1995). Use of a free radical method to evaluate antioxidant activity. https://doi.org/10.1016/S0023-6438(95)80008-5
As explained in the material and methods section (Lines 127-129), DPPH was expressed as the difference between blank and sample absorbance divided by blank absorbance × 100. For this reason, in some cases, DPPH radical scavenging activities below 20% were obtained whereas in the paper the reviewer suggested antiradical activity was measured as molar ratio of DPPH and antioxidant compound used, hence we do not think they can be compared. Nevertheless, since only one in vitro test cannot be descriptive of the whole antioxidant activity of a food sample, we performed the ABTS assay as well.
Reviewer 2 Report
The zero-waste approach is in scope of many research nowadays. There is a strong need to turn industry into more greener way. According to that the paper looks very interesting and promising. The application of side products of food industry has great potential mainly because of the possibility of functionalisation of wastes into valuable products. The idea of functionalisation and utilisation of grape pomace is also promising. The methodology of experiments planned is correct but needs some modification (see list below). The main question that needs to be stand is why authors omit the design of experiment. Application of e.g. RSM methodology will allow, in my opinion, to get more valuable information for general conclusion. But this is an question only because authors has planned experiments own way. The discussion corresponds with the data obtained. In my opinion the paper is almost ready for publication in Antioxidants. Some very minor revision (see list below) is needed to finally polish the paper. According to that I designate the paper as needed minor revision for the moment.
1. The variety of grapes should be indicated in methodology. It was green or dark/red grapes?
2. Please explain why the Gompertz model (with modification) was used in the experiment?
3. Does BHT was used as internal or external standard?
4. Please correct the data in table 4 “DPPH Radical Scavenging Activity (%) 7 ± 0.2” and other values. There is a precision mismatch. Should it be 7.0± 0.2?
Author Response
The zero-waste approach is in scope of many research nowadays. There is a strong need to turn industry into more greener way. According to that the paper looks very interesting and promising. The application of side products of food industry has great potential mainly because of the possibility of functionalisation of wastes into valuable products. The idea of functionalisation and utilisation of grape pomace is also promising. The methodology of experiments planned is correct but needs some modification (see list below). The main question that needs to be stand is why authors omit the design of experiment. Application of e.g. RSM methodology will allow, in my opinion, to get more valuable information for general conclusion. But this is an question only because authors has planned experiments own way. The discussion corresponds with the data obtained. In my opinion the paper is almost ready for publication in Antioxidants. Some very minor revision (see list below) is needed to finally polish the paper. According to that I designate the paper as needed minor revision for the moment.
The authors thank the reviewer for the comment.
Statistical approaches such as RSM are usually employed to maximize the production of a substance, optimizing operational factors. Indeed, RSM explores the relationship between several explanatory variables and one or more response variables. Although, ideally, this approach could have been used in this study, we believe it is not completely appropriate, nor the sole scope of the paper. We did not optimize microbial growth by evaluating the effect of different grape pomace, glucose, or yeast extract concentrations, also different pH values were not considered. On the contrary, we specifically selected precise parameters tath could enabled microbial growth which could not be obtained in the sole grape pomace. This approach was carried out solely to select the best performing starter, then used in the following experiments. The main focus of the paper was the study of the effect of fermentation, in terms of modification of the polyphenols profile and related antioxidant activity ex vivo, as well as the evaluation of fermentation as potential tool to up-cycle the pomace in a functional bread, without generating any more by-products.
Your suggestion will be surely taken into account in the further steps of the research, including the optimization of the pomace fermentation.
- The variety of grapes should be indicated in methodology. It was green or dark/red grapes?
OK, Paragraph 2.1 was implemented with info on grape cultivar (Line 74).
- Please explain why the Gompertz model (with modification) was used in the experiment? The Gompertz equation was modified by Zwietering to model microbial growth kinetics, however growth and acidification during fermentation often go at the same pace. Indeed, the sigmoidal shape of the pH profile during fermentation makes it suitable to be modelled using the modified Gompertz or similar equations (Nor-Khaizura et al. 2019. Modelling the effect of fermentation temperature and time on starter culture growth, acidification and firmness in made-in-transit yoghurt. LWT, 106, 113-121). Hence in this study, aiming at selecting the best adapting strain, we used the modified Gompertz equation to model acidification kinetics of the lactic acid bacteria used to ferment grape pomace. This aspect was clarified in Lines 110-111.
- Does BHT was used as internal or external standard?
BHT was used as reference synthetic antioxidant for the DPPH assay, which is a spectrophotometric assay hence it was used as positive control.
- Please correct the data in table 4 “DPPH Radical Scavenging Activity (%) 7 ± 0.2” and other values. There is a precision mismatch. Should it be 7.0± 0.2?
Decimals were updated in tables 2 and 4.